Microbiology
Spectrum

⊘ | **Open Peer Review** | Host-Microbial Interactions | Research Article

# Genetic architecture of heritable leaf microbes

Julia A. Boyle,[1] Megan E. Frederickson,[1] John R. Stinchcombe[1,2]

**ABSTRACT**   Host-associated microbiomes are shaped by both their environment and host genetics, and often impact host performance. The scale of host genetic variation important to microbes is largely unknown yet fundamental to the community assembly of host-associated microbiomes, with implications for the eco-evolutionary dynamics of microbes and hosts. Using *Ipomoea hederacea*, ivyleaf morning glory, we generated matrilines differing in quantitative genetic variation and leaf shape, which is controlled by a single Mendelian locus. We then investigated the relative roles of Mendelian and quantitative genetic variation in structuring the leaf microbiome and how these two sources of genetic variation contributed to microbe heritability. We found that despite large effects of the environment, both Mendelian and quantitative genetic host variation contribute to microbe heritability and that the cumulative small effect genomic differences due to matriline explained as much or more microbial variation than a single large effect Mendelian locus. Furthermore, our results are the first to suggest that leaf shape itself contributes to variation in the abundances of some phyllosphere microbes.

**IMPORTANCE**   We investigated how host genetic variation affects the assembly of *Ipomoea hederacea*'s natural microbiome. We found that the genetic architecture of leaf-associated microbiomes involves both quantitative genetic variation and Mendelian traits, with similar contributions to microbe heritability. The existence of Mendelian and quantitative genetic variation for host-associated microbes means that plant evolution at the leaf shape locus or other quantitative genetic loci has the potential to shape microbial abundance and community composition.

**KEYWORDS**   leaf shape, phyllosphere, *Ipomoea hederacea*, microclimate, host age, leaf microbiome, heritability, matriline, Mendelian, quantitative genetics

The community assembly of microbes on a host depends on the environment, the host's traits, and the microbes (1). Microbes that colonize leaves and other above-ground plant parts, known as the phyllosphere, can disperse to a leaf via air (2), rain (3), or soil (4), after which they experience selection due to conditions on the leaf surface and microbe-microbe interactions (1, 5). Host genetics play a role in the leaf microbiome: the similarity of leaf microbiomes across species depends on host phylogenetic relatedness (6, 7), and the similarity of microbiomes between individuals of the same species can depend on host intraspecific genetic variation (8–11). Here, we test how a single leaf shape gene affects leaf bacterial communities in *Ipomoea hederacea* and compare the magnitude of effects from that single locus to that of many small effect loci.

The genetically based physiological, morphological, or immune traits of hosts can lead to consistent associations between the abundance or community composition of microbes and host genotypes (12–15). When consistent host-microbe associations are present, the host's microbial phenotype can be considered a heritable host trait, even when the microbes are environmentally acquired (16–18). In other words, horizontally transmitted microbes can be heritable in host populations when host traits have predictable effects on microbial recruitment and alleles for these host traits are

Address correspondence to Julia A. Boyle, julia.boyle@mail.utoronto.ca.

The authors declare no conflict of interest.

See the funding table on p. 13.

transmitted from host parents to host offspring. The heritability of a host's microbial phenotype is important for understanding host evolution and ecology due to microbial effects on host fitness (19–21) and potential reciprocal selection between hosts and microbiomes (17, 22, 23). The variation in microbe heritability may also have interesting evolutionary repercussions for hosts, given that the phenotypic variance of many host traits can be microbially mediated (17, 18). Furthermore, if a host's microbial phenotype is due to heritable host genetic variation, the evolutionary forces affecting those host traits—selection, drift, mutation, migration, and non-random mating—can in turn affect microbial abundances, distributions, and microbiome composition (24). Like all heritable traits, host microbial phenotypes can be influenced by a few loci of large effect or the cumulative effects of many small effect loci, and understanding microbe heritability necessitates determining the scale of host variation that matters to microbes.

Mechanisms producing genotype-specific phyllosphere microbiomes are complex and varied. In *Arabidopsis thaliana*, a genome-wide association study of the phyllosphere microbial community found that loci related to defense and plant cell wall integrity affect microbial community variation, while species richness was affected by loci involved with viral reproduction, trichomes, and morphogenesis (9). Host genotypes may differ in immune genes and disease resistance, which have been linked to differences in switchgrass leaf fungal community structure (25) and inconsistent effects on bacterial and fungal maize phyllospheres (11). Plant genotypes can also vary in leaf morphology and other leaf attributes; for example, mutations in cuticle formation and ethylene production also affected the microbiome in a synthetic *A. thaliana* phyllosphere (26). The microclimate of a leaf is determined by many leaf properties, including surface temperature, surface area, thickness of the boundary layer of air, chemical composition, trichomes, gas exchange, and nutrients (27). The boundary layer of air is closest to the leaf surface and has warmer, humid air; non-segmented leaves with larger surface areas often have a thicker boundary layer, and thicker boundary layers can impede gas exchange from the leaves (28). Compounds and water released by leaves during gas exchange can be an important source of nutrients and resources for phyllosphere microbes (5), and they can impose selection on which microbial species can establish. These findings suggest that leaf morphology, and its effects on microclimates and resources, can potentially affect phyllosphere microbes.

Factors driving phyllosphere microbiome composition may depend strongly on the environment and the interaction between environment and genotype since the leaf microbiome is primarily environmentally acquired (29). Environmental factors like weather and seasonality influence which microbes are available to disperse to leaves and with what frequency (29). Phyllosphere microbes may differ over time due to changing moisture and temperature conditions abiotically filtering for microbes in the air and soil (30) and biotic interactions that emerge at different times, e.g., changing microbe-microbe interactions and new microbes from insects (31) or other plants (32). Determining the relative roles of environment, phenology, and plant genotype on the microbial community composition and abundance in the phyllosphere is an unresolved empirical challenge.

One unstudied potential source of host genotype influence on the leaf microbiome is leaf shape. Leaf shape is a key morphological difference between plant species (33) and could contribute to interspecific variation in leaf microbiomes. The shape of a leaf determines the temperature and humidity conditions it experiences and even its photosynthetic and gas exchange abilities (34, 35), potentially affecting microbial communities. Comparing the effects of leaf shape on the microbiome between plant species is confounded by other genetic and ecological differences between taxa, and for this reason, it is preferable to use a single species that exhibits intraspecific variation in leaf shape. *Ipomoea hederacea*, ivyleaf morning glory, is a flowering annual plant whose leaf shape is determined by a single Mendelian locus (36–38); homozygous genotypes are either fully lobed or whole, with heterozygotes being partially lobed (Fig. 1). In *I. hederacea,* homozygous lobed genotypes have been shown to be more protected

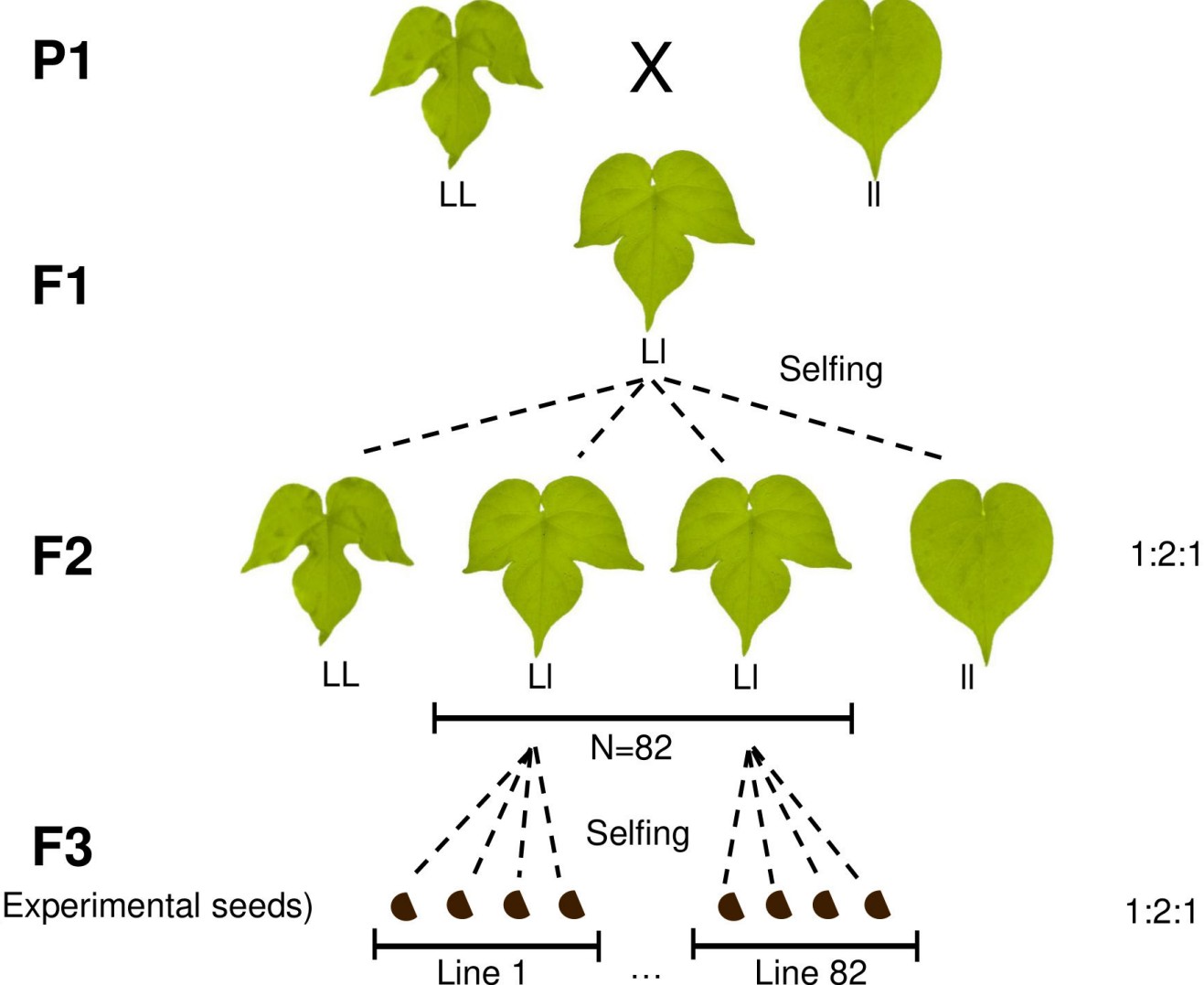

**FIG 1** *Ipomoea hederacea* leaf polymorphism and crossing design. Leaf shape is determined by a single Mendelian locus. First, both homozygous genotypes were collected from North Carolina, USA, and selfed for seven generations to create the parents (P1). To generate the plants used in the field experiment, the two homozygous P1 individuals were crossed, then heterozygote offspring were selfed for two more generations. Our experiment used F3 plants from 82 matrilines. Photos taken by Julia Boyle, with brightness and colors edited for style and clarity.

against extreme temperature changes compared to whole leaves; homozygous lobed leaves remained warmer at night by a mean difference of 0.16°C–0.22°C, likely due to their altered boundary layer resulting in leaves being more coupled with ambient air temperature (39, 40). The differences in *Ipomoea hederacea* leaf morphology and its effects on temperature and boundary layers that affect leaf microclimates suggest that different leaf shape genotypes could directly determine which colonizing microbes establish on the leaf, indirectly affecting microbe-microbe interactions and structuring leaf microbial communities as a whole.

To assess the relative contributions of quantitative and Mendelian genetic variation, we first examined if a single large effect Mendelian locus underlying leaf shape generates leaf microbiome differences within a host species. Then, we estimated the heritability of microbes due to many small effect loci that differ between plant lines—in other words, the amount of microbiome variation attributable to quantitative genetic causes. Finally, we compared the relative importance of host genotype to other factors such

as germination time, and therefore host age, to assess the role of the environment in structuring leaf microbiomes.

## RESULTS

To test the relative contributions of quantitative and Mendelian genetic variation in shaping *Ipomoea hederacea*'s leaf microbiome, we used a breeding design similar to that of recombinant inbred lines (Fig. 1). Our crosses created 82 matrilines differing in quantitative genetic variation due to recombination. Since the parents of the seeds used in the experiment were heterozygous, multiple leaf shape genotypes segregate within matrilines. The two temporal cohorts we used reflect the interlinked effects of plant age, developmental stage, and exposure to the environment on the leaf microbiome. After several months, we collected leaves from the 218 surviving plants and compared how germination cohort, leaf shape genotype, and matriline affected their leaf morphology, bacterial community, and the heritability of microbes.

### Genotype morphology

Leaf shape genotypes of *I. hederacea* differed quantitatively. Leaf shape genotypes differed significantly in measures of circularity [Wald $\chi^2_{(2,218)}$ = 1,520, $P < 0.001$; Table S2], with the whole genotype having the highest circularity followed by heterozygotes (Fig. S1). While there was a trend for leaf surface area to be the largest in whole leaves and lowest in the lobed leaves, the difference was not significantly predicted by leaf shape genotype [Wald $\chi^2_{(2,218)}$ = 4.15, $P > 0.05$; Table S2] or germination cohort [Wald $\chi^2_{(1,218)}$ = 2.22, $P > 0.05$; Table S2] (Fig. S1). As we specifically chose to sample similarly sized leaves, with the intent of minimizing differences in total leaf area, these results suggest our sampling strategy worked and that we can attribute differences in microbes due to leaf shape as not simply being the result of differences in leaf area.

### Community composition and diversity

We first characterized the microbial genera present on leaves and aspects of community composition and diversity. We considered a "core" microbial genus as one that is consistently associated with *I. hederacea* across temporal cohorts and in over 80% of leaves sampled. Across both cohorts, the core *I. hederacea* leaf microbes were the genera *Methylobacterium*, *Sphingomonas*, *Deinococcus*, *Pseudomonas*, *Arthrobacter*, and *Hymenobacter*. The older germination cohort had two additional microbial genera at above 80% prevalence across samples: *Frigoribacterium* and *Roseomonas*. Communities were structured by the high relative abundances of *Methylobacterium* and *Sphingomonas* genera (Fig. 2 and 3), and their relative abundances were negatively correlated (Fig. S2). The first principal coordinate analysis (PCoA) axis was strongly positively correlated with *Methylobacterium* relative abundance, while the second axis was strongly negatively correlated with *Sphingomonas* relative abundance (Fig. S2).

Germination cohort was the main driver of the overall microbial community composition. While microbiomes from both cohorts overlapped in composition, they were significantly different from each other [$F_{(1,182)}$ = 10.4, $P = 0.001$; Table S3]; older plants from the first cohort had more *Methylobacterium*, and younger plants from the second cohort had more *Sphingomonas* (Fig. 3A, C, and D). The younger cohort's microbiomes were more distinct from one another, creating significantly higher dispersion [$F_{(1,182)}$ = 10.4, $P = 0.001$; Table S3]. Germination cohorts always remained significantly different in composition and dispersion when cohort sample sizes were equalized (Fig. S3), suggesting this is a true biological phenomenon. Germination cohort significantly predicted observed richness [Wald $\chi^2_{(1,182)}$ = 10.1, $P = 0.001$; Table S4], with older plants having higher amplicon sequence variant richness than younger plants, but it did not predict Shannon diversity [Wald $\chi^2_{(1,182)}$ = 3.33, $P > 0.05$; Table S4] or evenness [Wald $\chi^2_{(1,182)}$ = 2.96, $P > 0.05$; Table S4].

Despite differences in morphology and microclimate that we *a priori* predicted to affect the leaf microbiome, we did not detect strong leaf shape effects on community

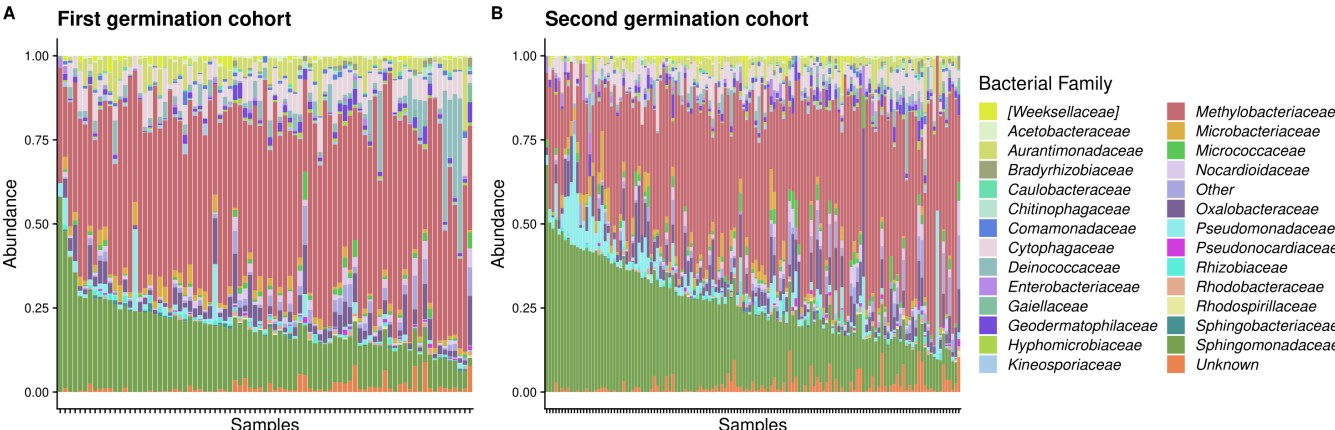

**FIG 2** Relative abundance of bacterial families in the *Ipomoea hederacea* leaf microbiome, indicated by color. Samples are ordered by *Sphingomonadaceae* abundance and grouped by germination cohort. The first germination cohort (A) was 89 days old ($n = 83$), and the second germination cohort (B) was 61 days old ($n = 135$) at the time of sampling. For this visualization, we included only families in at least 30% of samples and at least 1% relative abundance.

composition as a whole. Leaf shape did not significantly structure composition [$F_{(2,182)} = 1.83$, $P > 0.05$; Table S3] or affect Shannon diversity, evenness, or observed richness [Wald $\chi^2_{(2,182)} < 3.97$, $P > 0.05$ for all; Table S4]. While whole community composition was not affected, we nonetheless identified microbial genera significantly affected by leaf shape in the heritability analysis, as described below.

## Co-occurrence network analysis

Microbial networks capture information on microbial recruitment and environmental processes (41) and thus are a useful way to visualize (42) and characterize patterns of microbe niche similarity (43). To describe these patterns, we created co-occurrence networks among microbial genera for each leaf shape genotype and germination cohort. Microbial network properties varied between leaf shapes (Table S5). The homozygous lobe network had 122 significant correlations and the lowest percentage of positive correlations of all the networks, at only 51% (Fig. S4). The heterozygous network had 151 significant correlations (62% positive; Fig. S5), and the homozygous whole network had 68 significant correlations (63% positive; Fig. S6). Network modularity indicates distinct groupings (i.e., modules) of taxa that are highly correlated among themselves and less correlated with taxa in other groupings, while the number of clusters indicates how many distinct groupings of taxa were present. Modularity was highest in the homozygous whole network at 0.48, compared to 0.41–0.42 for the other genotypes' networks. Networks of leaf shape genotypes contained six to seven clusters. Germination cohorts had very similar overall microbial network properties to those of leaf shape; however, they formed only three to four clusters (Table S5; Fig. S7 and S8). The main difference between cohorts was the global clustering coefficient, which is a measure of how likely it is that the nodes correlated to a focal node are also correlated to each other; the older cohort had a coefficient of 0.24, while the younger cohort had a coefficient of 0.12. The top hub taxa were similar across all networks and tended to include *Methylobacterium*, *Sphingomonas*, and *Arthrobacter* (Table S5).

## Heritable microbes and community phenotypes

We next tested for genetic variation in microbial traits, first using community phenotypes and then the abundance of individual microbes. The community phenotypes tested included observed species richness (base 10 log-transformed), evenness, Shannon diversity, and the first three axes of the weighted UniFrac PCoA as the response variables in individual linear mixed-effects models. We calculated heritability as the ratio of matriline genetic variance ($V_G$) to the sum of matriline, spatial block, and residual

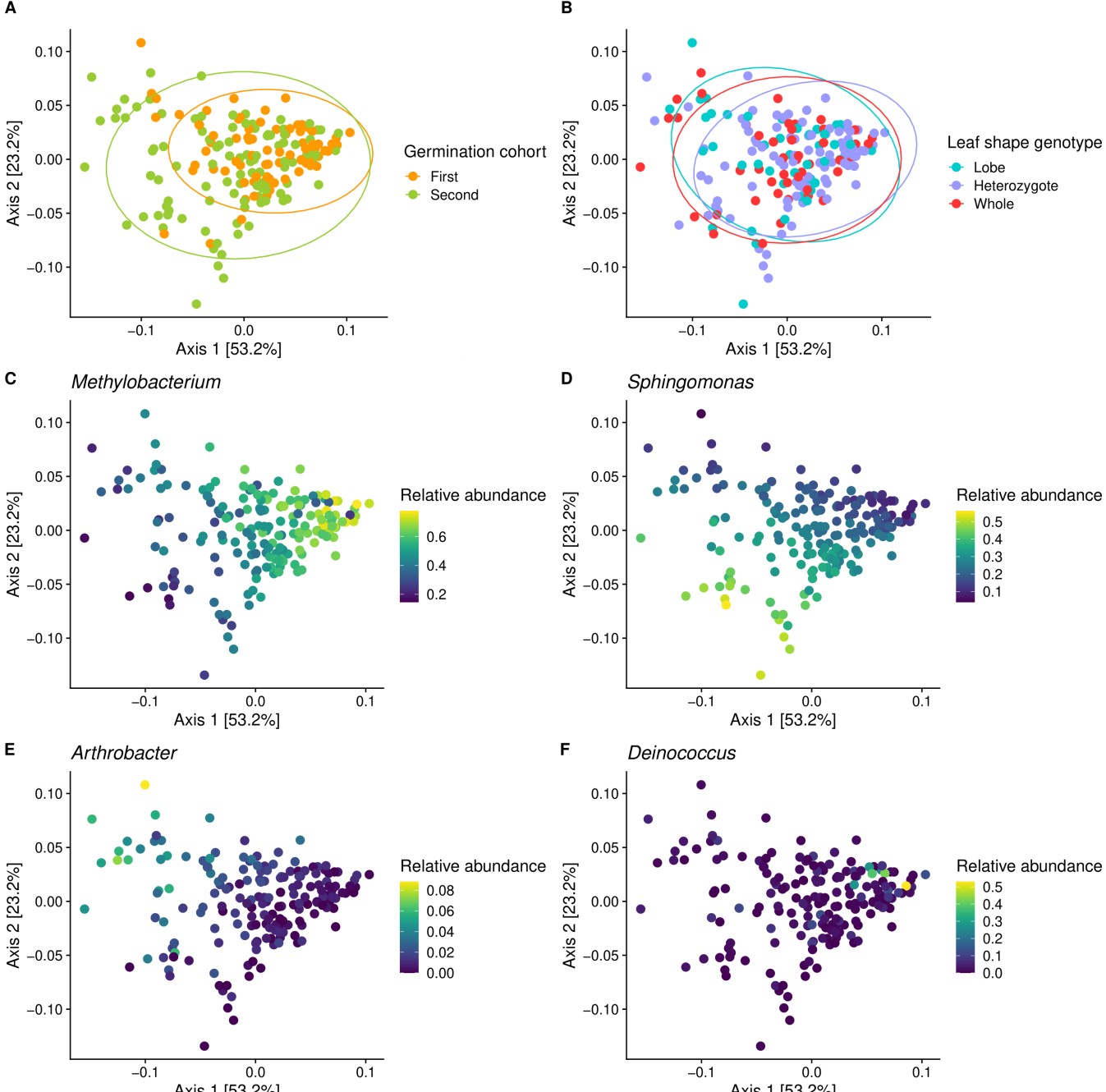

**FIG 3** Weighted UniFrac PCoA of *I. hederacea* leaf microbiomes. Germination cohort significantly predicted community composition (A), while leaf shape was non-significant (B). Core microbiota structured communities (C–F). For panels C–F, note that the relative abundance legend scales differ.

variance ($V_P = V_G + V_{Block} + V_E$). Community phenotypes had low broad-sense heritability ($0 \leq H^2 \leq {\sim}0.05$) with the third axis of the PCoA having the highest $H^2$, followed by Shannon diversity (Fig. 2A). All community phenotypes with non-zero $H^2$ were significantly heritable, as the models' log-likelihoods were improved by including matriline; community phenotypes with no $H^2$ (evenness and PCoA axis 2) showed no difference in fit when matriline was included (Fig. S9). The main effect of the germination cohort significantly affected observed richness, as previously described, as well as the first two axes of the PCoA [Wald $\chi^2_{(1,182)} > 6.71$, $P < 0.01$ for both axes; Table S4], whereas fixed effects of leaf shape did not significantly affect community phenotypes [Wald $\chi^2_{(2,182)} < 4.85$, $P > 0.05$ for all; Table S4]. In our qualitative comparison of the magnitudes of

variances explained, plant matriline had a larger effect on community phenotypes' $H^2$ than leaf shape, with the exception of observed richness (Fig. 4C; Table S6).

Host genetics had a stronger influence on the abundance of individual microbial genera than community phenotypes, with both quantitative genetic and Mendelian traits affecting microbe abundance. We considered only genera detected in at least 30% of samples and then used their center log-ratio-transformed abundance as responses in linear mixed-effects models, calculating heritability as before. We identified eight genera with non-zero and significant broad-sense heritability. The most heritable genera (0.05 ≤ $H^2$ ≤0.07) were *Nocardioides*, *Kineococcus*, *Pseudomonas*, and *Agrobacterium* (Fig. 4B). Likelihood ratio tests showed that including plant matriline improved model fit for 10 out of 15 genera (which included all genera with non-zero $H^2$); there was no difference in fit for three genera (*Deinococcus, Hymenobacter,* and *Sphingomonas*) and slightly reduced model fit for two genera (*Actinotelluria* and *Rubellimicrobium*) (Fig. S9). The germination cohort significantly affected the abundance of more than half of the most common genera [Wald $\chi^2_{(1,218)}$ > 3.83, $P$ < 0.05 for all; Table S7; Fig. 4B]. Four genera, *Sphingomonas*, *Nocardioides*, *Methylobacterium,* and *Agrobacterium,* had significant main effects of leaf shape on their phenotype [Wald $\chi^2_{(2,218)}$ > 7.03, $P$ < 0.05 for all; Table S7; Fig. 4B]. When we examined whether the magnitude of plant matriline variance was qualitatively comparable to leaf shape locus, we found that these four genera had equal to larger magnitudes of variance attributable to leaf shape than plant matriline (Fig. 4D; Table S8). In heritable genera with a non-significant main effect of leaf shape, plant matriline explained more variance than leaf shape (Fig. 4D; Table S8), as expected.

## DISCUSSION

Microbes can be considered an extended host phenotype with potentially adaptive functions for the host (44), and heritable microbes are more likely to consistently affect host phenotype and fitness through time. Here, we examined the relative contributions of Mendelian and quantitative genetic variation to the heritability of host-associated microbes to determine what types of host genetic variation mattered to microbial ecology. We found that both Mendelian and quantitative genetic host variation contribute to microbe heritability and that the cumulative small effect genomic differences due to matriline explained as much or more variation than a single large effect locus. We were able to identify heritable microbial variation despite a large effect of germination timing, which incorporated host age and the initial microbial environment, and was a major factor structuring community composition. We discuss the implications of our results below.

### Genetic influence on the microbiome

There was genetic variation at several scales mediating how well microbes can establish and perform in the phyllosphere. It is important to note that, as always, our estimates of heritability are specific to the population (45) and conditioned on fixed effects in the model (46), in our case, the germination cohort and leaf shape. A single leaf shape gene significantly explained the abundance of several common microbes and altered microbial network properties, suggesting the genetic differences in leaf microclimate (39, 40) (or other unknown features of the leaf shape genotypes affecting leaf attributes) impact microbial establishment and persistence and can contribute to intraspecific differences in the phyllosphere. Genetic differences between matrilines impacted microbes and could be related to loci unlinked to leaf shape, such as those related to plant immunity, metabolism, hormones, or other physical properties shaping the leaf microenvironment (9, 25, 26). When comparing the magnitude of effects, both leaf shape and matriline were important factors influencing microbe abundance, although plant matriline affected more of the microbes tested. Community phenotypes like species diversity and weighted UniFrac distances showed lower heritability than individual genera, but since the majority of microbes are not likely to be heritable in an

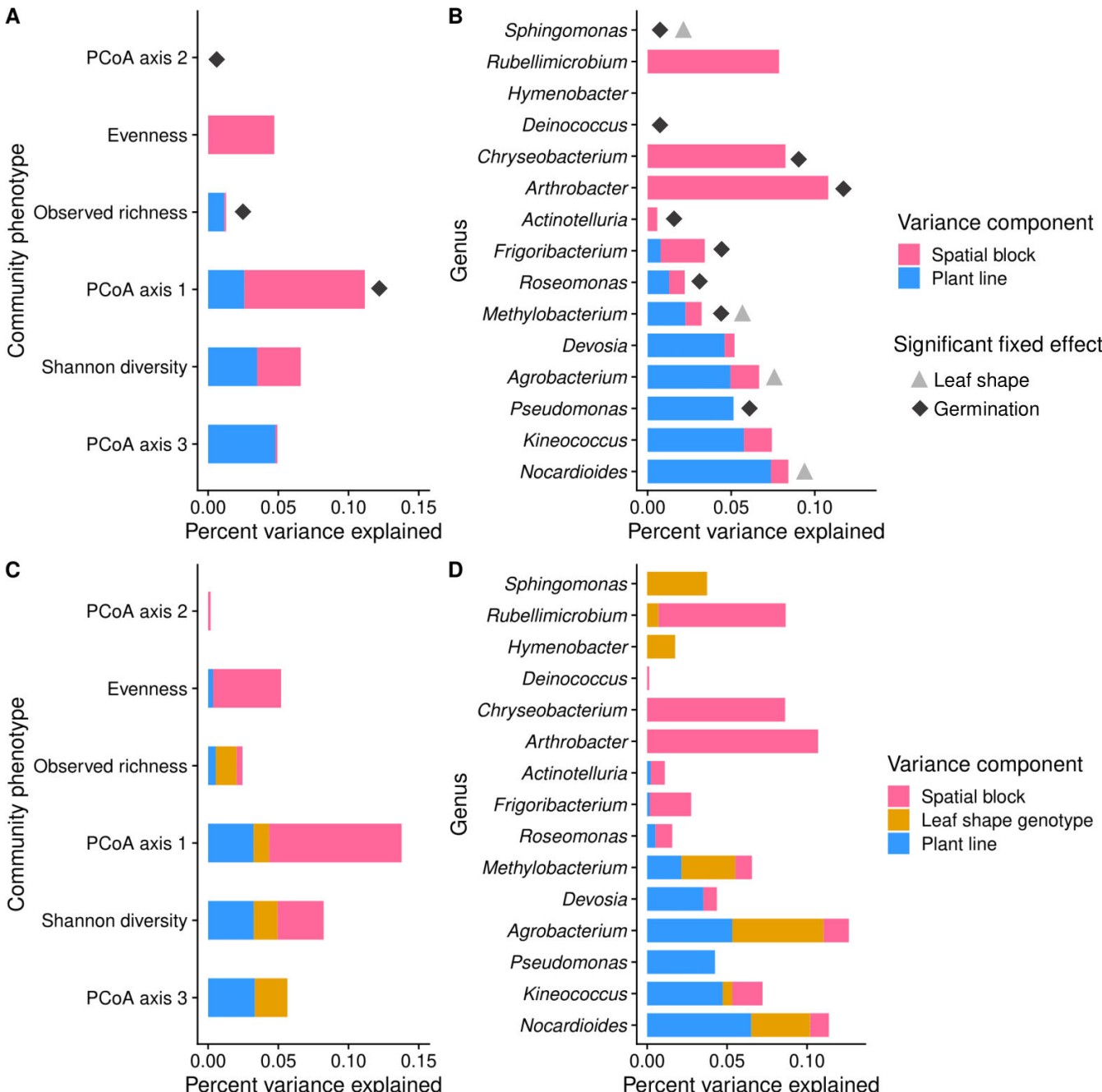

**FIG 4** Broad-sense heritability ($H^2$) of community phenotypes and common microbial genera. Genera are ordered by percent variance explained by plant matriline in the $H^2$ model. (A and B) $H^2$ of community phenotypes and genera in at least 30% of samples. $H^2$ was calculated as the percent variation explained by genetic line compared to the total of matriline, spatial block, and residual variances. Significant fixed effects of germination cohort and leaf shape are indicated by diamonds and triangles, respectively. (C and D) $H^2$ of community phenotypes and genera in at least 30% of samples, but with leaf shape as a random effect to directly compare the magnitude of variance explained to that of matriline.

environmentally acquired microbiome, very low heritability at a community scale was perhaps unsurprising. Our estimates of heritability are low to commensurate compared to other heritable microbiome studies (summarized in Table S9) (10, 12–14, 16, 47–51); while the range of heritabilities for plant-associated microbes across other studies was as broad as $H^2 = 0$–1, the average significant heritability of microbes and community phenotypes was usually low ($H^2 \leq 0.10$) (Table S9). Our results therefore follow the trend

of low heritability for environmentally acquired microbes, with a strong influence of the environment.

In our experiment, each plant line represented a different mosaic of the original parental genotypes, similar in concept to recombinant inbred lines (RILs). However, because the seeds we used in our experiment were F3 individuals, approximately 25% of the loci initially heterozygous in the F1 (i.e., those that differed between the parents) remain heterozygous within matrilines. One consequence of this is that there is genetic variation between matrilines due to the effects of recombination and genetic variation within matrilines (due to heterozygosity and segregation within a matriline). These effects make it more difficult to detect quantitative genetic variation in microbial phenotypes, especially compared to RIL populations that have been made homozygous within lines. Our results may thus underestimate the prevalence and magnitude of $H^2$ due to reduced power. Furthermore, our crossing design included only two parental genotypes, potentially limiting the amount of genetic variation segregating in the cross. For instance, had we sampled more genotypes to create MAGIC or nested association mapping lines (52, 53), more genetic variation would have been captured in our population. It is difficult to determine, *a priori*, whether this would lead to increased or decreased estimates of genetic variation. On the one hand, two parental genotypes are a limited sample of the genetic variation found in many populations. Although putatively neutral genetic variation in *I. hederacea* populations is often quite low (54), there is frequently quantitative genetic variation (55, 56). On the other hand, the selfing rate in *I. hederacea* is quite high (~93%) (54, 57). As such, rare outcrossing events, followed by selfing, may produce recombinant populations not that dissimilar from our experimental population. Similar arguments have been made for *A. thaliana* (58, 59), which is also highly selfing. As a qualitative investigation of the influence of including limited genetic variation in our crossing design, we gathered estimates of $H^2$ for life history and quantitative traits in *Arabidopsis thaliana*, specifically comparing studies that used either multiple accessions or RILs. These data (Table S10), while an imperfect comparison, suggest that if anything, our estimates of heritability could be under-estimated by using two parental lines in the initial cross.

## Evolutionary implications

We found similar magnitudes of variance in microbial phenotypes attributable to plant line and leaf shape genotype for several common bacterial genera, which is important because there is the potential for different evolutionary forces to act on quantitative genetic loci versus Mendelian loci. For example, Mendelian traits like leaf shape have a high potential to show sampling effects due to drift, especially in small populations (60). In contrast, drift is much less likely to produce a change in quantitative traits, which are influenced by many genes (60). Thus, drift can potentially lead to bigger effects on leaf-shape-associated microbes. Furthermore, high selfing in *I. hederacea* will reduce the frequency of heterozygous leaf shape genotypes by as much as 50% per generation. In this way, the mating system may have outsized effects on microbes linked to Mendelian traits. If natural selection acts on host traits, as it appears to with *I. hederacea*'s leaf shape (38), there may be indirect effects of selection on the microbial phenotype in the host population.

The microbes we identified as having heritable variation in abundance are also linked to host performance and fitness in other systems. These findings from other systems suggest intriguing hypotheses for further study. For example, one of the most heritable genera that was also significantly affected by leaf shape, *Agrobacterium*, has pathogens linked to causing tumor-like galls in plants (61), suggesting a potential link between leaf shape and disease risk. *Methylobacterium* spp. are linked to improving plant growth (62), and increased growth rate improves fitness in *I. hederacea* (39). *Pseudomonas* spp. have a wide range of positive or negative effects on plant performance (63). The most heritable genus *Nocardioides* is commonly endophytic on plant leaves and stems (64, 65) and has species capable of fixing nitrogen (66) and reducing nitrate (67). Evaluating whether any

of these microbial genera that have effects on host performance in other systems do so in *I. hederacea* will require further empirical work. If heritable microbes affect plant fitness, then there is a higher potential for reciprocal selection between the microbes and plant hosts.

## Environmental and phenological effects on the leaf microbiome

Community-wide differences between germination cohorts could be due to seasonal differences in what microbes were present in the plant's early life and the amount of time selection had to act on the microbial communities. The input of new microbes to the phyllosphere can be low relative to environments like soil rhizospheres, and these sources may change across the growing season (29). The cumulative input of microbes may matter as well since we found that the older plant phyllospheres had a higher species richness than younger plants; this is possibly because the leaves themselves were older and had more time to collect rare microbes dispersing from the environment. Additionally, as plants age and change phenologically, this may change the leaf environment for microbes (68–70). While in a time of growth and development, young plants have weaker immune responses (71), meaning they may impose weaker selection on their phyllospheres. Thus, the disparate microbiomes on young plants could be due to weaker selection from the environment and host over a shorter time. In contrast, older plants showed more similar and clustered microbiomes, suggesting that over time there is selection on microbes by the *I. hederacea* leaf microclimate, and the microbial community converges. The merging community compositions between young and old plants could suggest microbial succession occurred in the phyllosphere (72, 73). Early-season leaf microbiomes can be more influenced by soil microbes and become increasingly specific to the host plant over time (73); the core and highly abundant microbial genera in our samples are very common phyllosphere microbes (61–64, 73, 74), and thus do not suggest an outsized input of soil microbes.

The large effect of the germination cohort is not unexpected, given that most studies on microbiome heritability find a very large effect of environment and host age on the microbiome (10, 12, 16, 75). For example, Walters et al. (16) found that plant age was the largest driver of the maize rhizosphere, but in 1 year, they nonetheless found 143 heritable root microbes with $H^2$ = 0.15–0.25. While our results suggest genera like *Methylobacterium* and *Sphingomonas* were significantly affected by host age and initial microbial environment, host genotype still significantly mediated these environmental and phenological changes in microbe abundance. Our results add to recent evidence that the assembly process of host-associated microbiomes is governed by both stochastic forces and host-based selective forces (75, 76).

## Conclusions

Our results show that while *I. hederacea* leaf microbe composition differences were primarily shaped by host age and the environment, there exists a heritable subset of core microbes in the *I. hederacea* microbiome. A Mendelian trait and quantitative genetic variation across matrilines explained similar amounts of variation in microbial abundance, with implications for plant-microbe eco-evolutionary dynamics across time. Furthermore, we show for the first time that leaf shape itself may contribute to differences in phyllosphere microbial abundances within a species.

## MATERIALS AND METHODS

### Study system and crossing design

*Ipomoea hederacea*, ivyleaf morning glory, is a flowering annual plant, commonly found in eastern North America in roadside ditches and agricultural fields. It is predominantly selfing (77) and dies with the first hard frost in autumn. We used seeds derived from

a cross by Campitelli and Stinchcombe (39), where individuals from the two alternate homozygous leaf shape phenotypes (i.e., fully lobed or whole) were collected from North Carolina, USA, selfed for seven generations to generate homozygous parents (P1), and then crossed with each other (Fig. 1). A single F1 was allowed to self-fertilize, producing F2 progeny. We scored F2 plants for leaf shape and allowed them to self-fertilize; we refer to all the selfed progenies of an individual F2 plant as a "matriline." We used F3 seeds, set by F2 plants we had identified as heterozygous for leaf shape, as our experimental seeds. In total, we used seeds from 82 matrilines. As expected, genotype frequencies did not significantly differ from a 1:2:1 ratio for leaf shape [$\chi^2_{(2, 218)}$ = 3, $P$ = 0.22], with 55 plants with whole leaves, 112 plants with partially lobed leaves, and 51 plants with fully lobed leaves.

Our breeding design has three consequences for leaf microbiomes, which we address in the Discussion. First, variation between matrilines is due to the combined effects of loci that differed between the original parents of the cross and the effects of recombination. Second, because of the effects of recombination, significant differences between leaf shape genotypes are due to either the effects of the leaf shape locus itself or linked loci not broken up by recombination. Third, within a matriline, we expect 25% of the loci that differed between the parents to still be heterozygous, which is considerably more within-matriline genetic variation than would be found in inbred lines or RILs, potentially reducing our power and making our estimates of microbe heritability more conservative.

## Field site and experimental design

In 2021, we planted a total of 240 *I. hederacea* seeds from 82 matrilines (2–3 seeds/matriline) in a common garden at Koffler Scientific Reserve (https://ksr.utoronto.ca/) in Ontario, Canada. We scarified seeds, then planted them in a greenhouse in peat pots containing soil from the field. We planted the first cohort on 4 June and a second cohort on 2 July, because of poor germination in the first cohort. After the majority of plants in a given cohort germinated (approximately 1 week), we transplanted the pots into the field. We plowed and disked an old field for the common garden site and placed plants 1 m apart with a 6 ft tall bamboo pole to climb. We used three spatial blocks in the field, with one replicate per matriline found within each spatial block; within blocks, we randomly assigned positions of each matriline. The common garden was moderately weeded until the *I. hederacea* plants had established; neighboring plants in the common garden were predominantly *Cirsium arvense*, Canada thistle. The 218 surviving plants consisted of 83 individuals from the first cohort and 135 individuals from the second cohort. There were no significant differences in the frequencies of leaf shape genotypes between the two cohorts [$\chi^2_{(2, 218)}$ = 0.31, $P$ = 0.86; Table S1]. On 1 September, we collected one leaf within 1 foot from the ground from each plant by cutting the petiole with sterilized scissors into a sterile plastic bag, after which leaves were stored in a −80°C freezer until DNA extraction. We avoided collecting atypical leaves, for example, leaves with herbivory damage, sun damage, small and large leaves, or leaves that looked unusually dirty. The soil is a known contributor to leaf microbiomes (4, 73), and while it is likely that soil microbes interacted with the leaf surface through rain and wind, we expect that our collection methods led to a predominance of leaf microbes. At the time of leaf collection, plants in the first germination cohort were 89 days old with 96% of plants flowering, and plants in the second germination cohort were 61 days old with 49% of plants flowering. As a consequence, the second cohort of plants were younger, experienced field conditions for fewer days, and were developmentally younger. We photographed collected leaves against a 1 cm$^2$ grid background and measured leaf surface area and shape using ImageJ (78). From these photographs, we calculated leaf circularity as 4π (area/perimeter$^2$), which has a value between 0 and 1, with 1 indicating a perfect circle.

## Sequencing and QIIME2 analysis

We performed extractions using the whole leaves with QIAGEN DNeasy PowerSoil Pro Kits; both epiphytic and endophytic microbes were extracted. We sent samples to Génome Québec (Montréal, Canada) for Illumina MiSeq PE 250 bp 16S rRNA gene amplicon sequencing on the conserved hypervariable V4 region (primer pair 515F-806R). We used Quantitative Insights Into Microbial Ecology 2 (QIIME2) v.2022.2 (79) to trim the sequences for quality, and we denoised the sequences with DADA2 (80). Samples had a median read frequency of 35,436 reads. Using QIIME2, we removed amplicon sequence variants (ASVs) that had fewer than 10 reads across all samples and assigned taxonomy using the "sklearn" feature classifier and Greengenes 16S rRNA gene V4 region reference (81), then filtered out reads assigned as cyanobacteria and mitochondria to remove plant DNA. After these steps, we had 4,934 ASVs across the 218 leaf samples, with samples having a median read frequency of 4,523 reads. After visualizing a rarefaction curve, we created a rarefied data set with 4,695 ASVs total across 183 samples, with 1,809 reads/sample; the first and second cohorts retained 77 and 106 samples, respectively. Finally, we constructed a phylogeny using QIIME2's MAFFT (82) and FastTree 2 (83) to obtain a rooted tree.

## Statistical analysis

For statistical analysis, we used R v4.2.0 (84), with the tidyverse (85), lmerTest (86), phyloseq (87), vegan (88), and microbiome (89) packages. The general model structure we used included leaf shape genotype and germination cohort as fixed effects and random effects of block and matriline unless otherwise specified. All linear models were adjusted with type III ANOVAs, calculated in the car package (90).

First, we compared leaf shape surface area and circularity between genotypes using linear mixed-effects models. Next, we identified the core microbes by filtering the rarefied data set to genera found in at least 80% of samples across both cohorts. We also compared genera that passed this 80% prevalence threshold in each germination cohort separately. To assess the effect of leaf shape and germination cohort on microbial composition, we calculated a weighted UniFrac distance matrix using rarefied data, then we used an adonis2 PERMANOVA (999 permutations) followed by a permutational test of dispersion on significant groups using betadisper. The vegan package does not allow for random effects, so we did not include block and matriline. To confirm that germination cohort sample sizes did not affect our results, we also downsampled to equal cohort sizes 100 times and used the same permutation tests for composition and dispersion (Fig. S3). To visualize the community, we used a weighted UniFrac principal coordinate analysis using rarefied relative abundance data. We created microbial co-occurrence networks at the genus level for each leaf shape genotype and germination cohort. Using SparCC and sparccboot implemented through the SpiecEasi package (91), we permuted the correlations 100 times to generate a null expectation, then calculated false-discovery rate adjusted $P$ values for microbial pairs. Using the networks of only significantly correlated genera, we then calculated modularity, Kleinburg's hub centrality scores, and the global clustering coefficient.

We estimated broad-sense heritability ($H^2$) of microbial community phenotypes and taxonomic groups using linear mixed-effects models, taking a similar approach as Wagner et al. (10) and Grieneisen et al. (14). Here, the random effects of block and matriline explain variance in the microbe abundance phenotype after accounting for the mean effects of the germination cohort and leaf shape, meaning our estimates of $H^2$ are conditioned on the fixed effects of the germination cohort and leaf shape (as is commonly the case [46]). To estimate $H^2$ from the random effects, we calculated the ratio of matriline genetic variance ($V_G$) to the sum of matriline, spatial block, and residual variance ($V_P = V_G + V_{Block} + V_E$). We generated the community phenotypes using rarefied data and included observed species richness (base 10 log-transformed), evenness, Shannon diversity, and the first three axes of the weighted UniFrac PCoA as the

response variables in individual linear mixed-effects models. To estimate the heritability of taxonomic groups in a more compositionally aware way (92), we aggregated the non-rarefied data set to genera that were in at least 30% of samples, added one to each abundance, and center log-ratio transformed the abundance matrix. Next, we fit a linear mixed-effects model to the abundance of each genus that was in at least 30% of samples. To assess the statistical significance of the matriline effect, we compared the log-likelihood of our models with and without line as a random effect; the difference in log-likelihoods is $\chi^2$ distributed, with one degree of freedom (93). Finally, solely for the purposes of a qualitative comparison of the magnitude of variance explained by the leaf shape locus, matriline effects, spatial blocks, and residual variation, we used the same mixed-effects model except with leaf shape as a random effect. The data sets analyzed during the current study are available in the Dryad repository, https://doi.org/10.5061/dryad.rbnzs7hjz.

## ACKNOWLEDGMENTS

We thank Brandon Campitelli for generating the morning glory lines and the staff of Koffler Scientific Reserve in 2021 for helping to maintain the field site. We acknowledge Génome Québec (Montréal, Canada) for sequencing samples. Finally, we gratefully acknowledge the Swedish Collegium for Advanced Study, and Jennifer Mack, Natuschka Lee, Iva Lučić, and Arthur Asseraf for their support.

Funding sources include NSERC graduate funding to J.A.B. and NSERC discovery grants to M.E.F. and J.R.S.

## AUTHOR AFFILIATIONS

[1]Department of Ecology and Evolutionary Biology, University of Toronto, Toronto, Ontario, Canada
[2]Swedish Collegium for Advanced Study, Uppsala, Sweden

## AUTHOR ORCIDs

Julia A. Boyle http://orcid.org/0000-0003-1029-8341
Megan E. Frederickson http://orcid.org/0000-0002-9058-7137
John R. Stinchcombe http://orcid.org/0000-0003-3349-2964

## FUNDING

| Funder | Grant(s) | Author(s) |
| --- | --- | --- |
| Canadian Government | Natural Sciences and Engineering Research Council of Canada (NSERC) | | Megan. E. Frederickson |
| | | John R. Stinchcombe |
| | | Julia A. Boyle |

## AUTHOR CONTRIBUTIONS

Julia A. Boyle, Conceptualization, Formal analysis, Investigation, Methodology, Visualization, Writing – original draft, Writing – review and editing | Megan E. Frederickson, Formal analysis, Funding acquisition, Investigation, Methodology, Supervision, Visualization, Writing – original draft, Writing – review and editing | John R. Stinchcombe, Conceptualization, Formal analysis, Funding acquisition, Investigation, Methodology, Supervision, Visualization, Writing – original draft, Writing – review and editing

## DATA AVAILABILITY

The data sets generated and code used to analyze them are available in the Dryad repository (https://doi.org/10.5061/dryad.rbnzs7hjz) and on Github (https://github.com/

JuliaBoyle/ipomoea_microbiome). Raw microbiome sequence data are available at NCBI's Sequence Read Archive under the BioProject accession number PRJNA1107536.

## ADDITIONAL FILES

The following material is available online.

### Supplemental Material

**Supplemental figures (Spectrum00610-24-s0001.docx).** Fig. S1-S9.
**Supplemental tables (Spectrum00610-24-s0002.docx).** Tables S1-S10.

### Open Peer Review

**PEER REVIEW HISTORY (review-history.pdf).** An accounting of the reviewer comments and feedback.

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
