## [Reviewer comments · Microbiology Spectrum]

Microbiology Spectrum

Genetic architecture of heritable leaf microbes

Julia Boyle, Megan Frederickson, and John Stinchcombe

Corresponding Author(s): Julia Boyle, University of Toronto

Review Timeline:

Submission Date:	March 6, 2024
Editorial Decision:	April 10, 2024
Revision Received:	April 22, 2024
Accepted:	May 2, 2024

Editor: Xue Zhang

Reviewer(s): The reviewers have opted to remain anonymous.

Transaction Report:

DOI: <https://doi.org/10.1128/spectrum.00610-24>

Re: Spectrum00610-24 (Genetic architecture of heritable leaf microbes)

Dear Dr. Julia A Boyle:

Thank you for the privilege of reviewing your work. Below you will find my comments, instructions from the Spectrum editorial office, and the reviewer comments.

Revision Guidelines

Sincerely,
Xue Zhang
Editor
Microbiology Spectrum

Reviewer #1 (Comments for the Author):

The manuscript "Genetic architecture of heritable leaf microbes" presents a detailed investigation into how genetic variation within the host plant *Ipomoea hederacea* influences the assembly and composition of its leaf microbiome. The study effectively combines a well-designed experimental setup with comprehensive statistical analysis to explore the relative contributions of Mendelian and quantitative genetic variation to microbe heritability. The findings offer significant insights into the eco-evolutionary dynamics of plant-microbe interactions and the role of genetic architecture in shaping microbial communities on

plant leaves.

Major comments:

- 1 . The authors could describe in greater detail the methods used to minimize or measure environmental variability within the experimental setup. For instance, specifying whether environmental conditions (e.g., temperature, humidity, sunlight exposure) were monitored and controlled or how spatial blocking was employed to reduce environmental heterogeneity.
- 2 . The manuscript could benefit from a discussion section that speculates on possible mechanisms through which genetic variations, particularly those influencing leaf shape, might affect the microbiome. This could include referencing literature on how leaf morphology impacts microclimate, leaf surface properties, or plant metabolic profiles that in turn influence microbial colonization and survival.
- 3 . The figures need to be improved and provide the reproducible script in GitHub. Such as ImageGP (<https://doi.org/10.1002/imt2.5>) can generate high quality figures and with reproducible scripts.
- 4 . The structure of the results appears not in concise. The authors should have presented their findings in around 3 mainly sections will be better. For example, Figure 1 & 2 can be one, because a design of Figure 1 just as a panel A is enough.
- 5 . For statistical analysis, EasyAmplicon have better reproducible steps and scripts is recommended. A network compare with each group is also recommended, such as ggClusterNet, iNAP are good tools for network analysis and visualization.

Minor comments:

1. Pay attention to formatting, such as standardizing spaces around symbols (e.g., "=", "<", ">")
2. Line 68: is important to→is important for

Reviewer #2 (Public repository details (Required)):

They provided a link to view the files temporarily, but the other two links are not working.

Reviewer #2 (Comments for the Author):

The authors utilized the Ipomoeae hederacea leaf shape locus, to study its correlation with the microbiome. The paper is well-written, and the discussion aligns with the results obtained. The authors have also acknowledged the limitations of microbes and host correlations. The study shows that specific microbes are linked to leaf shape, rather than the entire microbial community.

I have some questions that were not clear in the paper:

- The methodology did not explain how the authors define the core microbiome. Could there be different cores for each germination cohort?
- How do the authors consider the different developmental stages of the two collected cohorts?
- How likely is it that leaves might get contaminated from soil microbiome?
- From the methodology, it is unclear for each seed they planted if they collected multiple or single leaves and if that represents a single sample.

Genetic architecture of heritable leaf microbes

Julia A. Boyle, Megan. E. Frederickson, John R. Stinchcombe

RESPONSE TO EDITOR

Thank you Dr. Zhang for overseeing this manuscript. In response to reviewer comments we clarified our methods and added more detail in the discussion. As reviewer #1 suggested, we added a co-occurrence network analysis to compare networks between leaf shape genotypes and germination cohorts. This analysis supports that leaf shape genotypes differ in a way that impacts microbes. The manuscript has improved thanks to the reviewers' feedback.

RESPONSE TO REVIEWER #1

COMMENT: The manuscript "Genetic architecture of heritable leaf microbes" presents a detailed investigation into how genetic variation within the host plant *Ipomoea hederacea* influences the assembly and composition of its leaf microbiome. The study effectively combines a well-designed experimental setup with comprehensive statistical analysis to explore the relative contributions of Mendelian and quantitative genetic variation to microbe heritability. The findings offer significant insights into the eco-evolutionary dynamics of plant-microbe interactions and the role of genetic architecture in shaping microbial communities on plant leaves.

RESPONSE: We thank the reviewer for their feedback!

COMMENT: 1. The authors could describe in greater detail the methods used to minimize or measure environmental variability within the experimental setup. For instance, specifying whether environmental conditions (e.g., temperature, humidity, sunlight exposure) were monitored and controlled or how spatial blocking was employed to reduce environmental heterogeneity.

RESPONSE: We did not monitor or control the environmental conditions at our field site, but we randomized matriline within each spatial block so leaf shape and genotype were not confounded with environmental variation. We added more detail on our within-block randomization to further clarify the experimental design in the methods.

Now it reads [Line 438]: "We plowed and disked an old field for the common garden site. We used 3 spatial blocks in the field, with 1 replicate per matriline found within each spatial block; within blocks, positions of each matriline were randomly assigned. "

COMMENT: 2. The manuscript could benefit from a discussion section that speculates on possible mechanisms through which genetic variations, particularly those influencing leaf shape, might affect the microbiome. This could include referencing literature on how leaf morphology impacts microclimate, leaf surface properties, or plant metabolic profiles that in turn influence microbial colonization and survival.

RESPONSE: We now reference the literature more in our discussion of how leaf shape and genetic variation may affect microbes.

The discussion [Lines 296] now reads: “A single leaf shape gene significantly explained the abundance of several common microbes and altered microbial network properties, suggesting the genetic differences in leaf microclimate [39, 40] (or other, unknown features of the leaf shape genotypes affecting leaf attributes) impact microbial establishment and persistence, and can contribute to intraspecific differences in the phyllosphere. Genetic differences between matriline impacted microbes, and could be related to loci unlinked to leaf shape, such as those related to plant immunity, metabolism, hormones, or other physical properties shaping leaf microenvironment [9, 25, 26].”

COMMENT: 3. The figures need to be improved and provide the reproducible script in GitHub. Such as ImageGP (<https://doi.org/10.1002/imt2.5>) can generate high quality figures and with reproducible scripts.

RESPONSE: In our Dryad data repository (peer-reviewer link: <https://datadryad.org/stash/share/haVENGR8LbK8GAnCP5CdrpBoqPQuTNc0IK1GCrUxBdk>), we provided reproducible code in an R markdown file. This file contains all the code required to run the analyses and to produce the figures containing results. The code will be available permanently both in the Dryad repository (<https://doi.org/10.5061/dryad.rbnzs7hjz>) and on Github (https://github.com/JuliaBoyle/ipomoea_microbiome) upon publication.

COMMENT: 4. The structure of the results appears not in concise. The authors should have presented their findings in around 3 mainly sections will be better. For example, Figure 1 & 2 can be one, because a design of Figure 1 just as a panel A is enough.

RESPONSE: We appreciate the reviewer’s feedback. The goal of Figure 1 is to show the crossing design used to generate matriline (methods), while Figure 2 presents the microbial community composition of germination cohorts (unrelated results to the methods in Figure 1). Because the goals of Figure 1 and Figure 2 are so different, we have elected to retain them as separate figures.

COMMENT: 5. For statistical analysis, EasyAmplicon have better reproducible steps and scripts is recommended. A network compare with each group is also recommended, such as ggClusterNet, iNAP are good tools for network analysis and visualization.

RESPONSE: We appreciate the reviewer's suggestion and will consider it in future projects. We used QIIME2 and R for our analysis. QIIME2 files retain provenance, which outlines the steps taken to produce the files; we also describe these steps in detail in our methods under the 'Sequencing and QIIME2 analysis' heading. All subsequent analyses using the QIIME2 feature table are in our R markdown file, which contains all code needed to reproduce the analysis.

We agree that co-occurrence network analyses provide another interesting metric of microbe niche similarity and microbe-microbe interactions, and now cite ggClusterNet as a tool to visualize these patterns [Line 167]. We used SparCC to create networks of significant microbial correlations for the three different leaf shapes, as well as the two germination cohorts. We found that homozygous lobe leaves have a higher proportion of negative correlations compared to other networks for leaf shape and cohort. Modularity was highest in homozygous heart shaped leaves. Top hub genera were similar across most of the networks. This new analysis is included in the methods [Line 496] and results [Line 165]. The summary table and new network figures are included in the supplemental material.

COMMENT: 1. Pay attention to formatting, such as standardizing spaces around symbols (e.g., "=", "<", ">")

RESPONSE: We have standardized spacing around symbols throughout the manuscript.

COMMENT: 2. Line 68: is important to→is important for

RESPONSE: We have made the suggested change.

RESPONSE TO REVIEWER #2

COMMENT: They provided a link to view the files temporarily, but the other two links are not working.

RESPONSE: We provided a private-for-peer-review link to the Dryad repository for temporary access (peer-reviewer link:

<https://datadryad.org/stash/share/haVENGR8LbK8GAnCP5CdrpBoqPQuTNc0IK1GCrUxBdk>). The other links (permanent Dryad link and github link) were set as private during the peer review process (<https://doi.org/10.5061/dryad.rbnzs7hjz> and https://github.com/JuliaBoyle/ipomoea_microbiome). Public access to these permanent repositories will be granted upon acceptance for publication.

COMMENT: The authors utilized the *Ipomoeae hederacea* leaf shape locus, to study its correlation with the microbiome. The paper is well-written, and the discussion aligns with the results obtained. The authors have also acknowledged the limitations of microbes and host correlations. The study shows that specific microbes are linked to leaf shape, rather than the entire microbial community.

RESPONSE: We thank the reviewer for their time and feedback!

COMMENT: The methodology did not explain how the authors define the core microbiome. Could there be different cores for each germination cohort?

RESPONSE: In the methods, we stated that “we identified the core microbiota by filtering the rarefied dataset to genera found in at least 80% of samples” [Line 485]. Therefore, we considered a ‘core’ microbe as one that is consistently associated with *Ipomoea hederacea* across temporal cohorts, using an occurrence cutoff that is common in the literature (Neu et al. 2021).

To address the reviewer’s question, we tested whether germination cohorts have different core microbes. Across both germination cohorts, core microbes were *Arthrobacter*, *Deinococcus*, *Hymenobacter*, *Methylobacterium*, *Pseudomonas*, and *Sphingomonas*. The older germination cohort had two additional microbial genera at >80% prevalence across samples: *Frigoribacterium* and *Roseomonas*. We updated the results [Line 192] and methods [Line 486] to reflect this additional information.

COMMENT: How do the authors consider the different developmental stages of the two collected cohorts?

RESPONSE: We added additional text to the manuscript to communicate how the two cohorts differ both in age/developmental stage and initial microbial environment.

In the methods [Line 453] we now state “At the time of leaf collection, plants in the first germination cohort were 89 days old with 96% of plants flowering, and plants in the second germination cohort were 61 days old with 49% of plants flowering. As a consequence, the second cohort of plants were younger, experienced field conditions for fewer days, and were developmentally younger.”

We added more explicit language at the beginning of the results [Line 138]: “The two temporal cohorts we used reflect the interlinked effects of plant age, developmental stage, and exposure to the environment on the leaf microbiome.”

We have elected to retain using cohort as a predictor, rather than developmental stage or flowering status, as it captures all of these effects simultaneously. Importantly, cohort status captures time of exposure to the environment in a way not captured by flowering status (i.e., second cohort individuals that have flowered by the time of sampling were still exposed to the environment for fewer days). For all the analyses presented in the paper, we include cohort as a predictor in all statistical models.

When we explored using flowering status as a predictor, we found that does not change the main results. The linear model results testing alpha diversity are unchanged, as well as the community composition results from the PERMANOVA. The models for the heritability of community phenotypes remain qualitatively unchanged both in significance and heritability estimates. Heritability estimates of genera remain very similar, and all genera significant for leaf shape remain so. Genera significantly affected by germination cohort remain the same, or become marginally significant; only one genus, *Roseomonas*, is significantly affected by flowering status ($X^2=4.20$, $p=0.040$) with a marginally significant effect of germination cohort ($X^2=3.67$, $p=0.055$).

Given the similar results between these analyses, and that cohort status also captures temporal exposure, we have elected to retain the original in the main text.

COMMENT: How likely is it that leaves might get contaminated from soil microbiome?

RESPONSE: We collected leaves from plants in the field, about 1 foot above the ground, making it very likely that soil microbes interacted with the leaf surface through rain, wind, etc. However, the soil microbiome is a known contributor to leaf microbiomes (one example is Copeland et al. 2015), and plants in the wild

interface with soil microbes very frequently, making the microbiomes we collected ecologically realistic.

We have now added more detail in the methods [Line 449] to comment on some of these factors: “We avoided collecting atypical leaves, for example, leaves with herbivory damage, sun damage, small and large leaves, or leaves that looked unusually dirty. The soil is a known contributor to leaf microbiomes [4, 73], and while it is likely that soil microbes interacted with the leaf surface through rain and wind, we expect that our collection methods led to a predominance of leaf microbes.”

Additionally, the microbiomes are overwhelmingly composed of *Methylobacterium* and *Sphingomonas* (Figure 2) which are very common phyllosphere microbes (Lee et al. 2006; Luo et al. 2019). Many of the core microbiota are also commonly found associated with plants (Lippincott & Lippincott 1969; Copeland et al. 2015; Kämpfer et al. 2016; Wang et al. 2022). The composition of the microbiome therefore does not suggest an outsized input of soil microbes.

We have added the following to the discussion [Line 385]: “Early-season leaf microbiomes can be more influenced by soil microbes and become increasingly specific to the host plant over time [73]; the core and highly abundant microbial genera in our samples are very common phyllosphere microbes [61-64, 73, 74], and thus do not suggest an outsized input of soil microbes.”

COMMENT: From the methodology, it is unclear for each seed they planted if they collected multiple or single leaves and if that represents a single sample.

RESPONSE: We thank the reviewer for pointing this out. We clarified in the methods that we collected a single leaf per plant.

Now it reads [Line 446]: “On September 1st, we collected one leaf within one foot from the ground from each plant by cutting the petiole with sterilized scissors into a sterile plastic bag, after which leaves were stored in a -80°C freezer until DNA extraction.”

Literature Referenced

Copeland J, et al. Seasonal Community Succession of the Phyllosphere Microbiome. *PMI*. 2015;28:274-285.

Lippincott JA, Lippincott BB. Tumour-initiating Ability and Nutrition in the Genus *Agrobacterium*. J Gen Microbiol. 1969;59:57–75.

Lee HS, et al. Physiological enhancement of early growth of rice seedlings (*Oryza sativa* L.) by production of phytohormone of N₂-fixing methylotrophic isolates. Biol Fertil Soils. 2006;42:402–8.

Luo L, et al. Variations in phyllosphere microbial community along with the development of angular leaf-spot of cucumber. AMB Expr. 2019;9:76.

Neu A, et al. Defining and quantifying the core microbiome: Challenges and prospects. PNAS. 2021;118:e2104429118.

Wang NR, et al. Commensal *Pseudomonas fluorescens* Strains Protect *Arabidopsis* from Closely Related *Pseudomonas* Pathogens in a Colonization-Dependent Manner. MBio. 2022;13:e02892-21.

Kämpfer P, et al. *Nocardioides zeicaulis* sp. nov., an endophyte actinobacterium of maize. Int J Syst Evol Microbiol. 2016;66:1869–74.

Re: Spectrum00610-24R1 (Genetic architecture of heritable leaf microbes)

Dear Dr. Julia A Boyle:

Your manuscript has been accepted, and I am forwarding it to the ASM production staff for publication. Your paper will first be checked to make sure all elements meet the technical requirements. ASM staff will contact you if anything needs to be revised before copyediting and production can begin. Otherwise, you will be notified when your proofs are ready to be viewed.

Sincerely,
Xue Zhang
Editor
Microbiology Spectrum

Reviewer #1 (Comments for the Author):

The author's response has been fully addressed my concerns. The quality of the paper has apparently improved. I agree with the publication of this article.

Reviewer #2 (Comments for the Author):

The authors have followed the reviewers' suggestions and I have no further comments.